# Genomic Insights into the Symbiotic and Plant Growth-Promoting Traits of “*Candidatus Phyllobacterium onerii*” sp. nov. Isolated from Endemic *Astragalus flavescens*

**DOI:** 10.3390/microorganisms12020336

**Published:** 2024-02-06

**Authors:** Asiye Esra Eren Eroğlu, Volkan Eroğlu, İhsan Yaşa

**Affiliations:** 1Basic and Industrial Microbiology Section, Biology Department, Faculty of Science, Ege University, 35100 Izmir, Türkiye; asiye.esra.eren@ege.edu.tr; 2Botany Section, Biology Department, Faculty of Science, Ege University, 35100 Izmir, Türkiye; volkan.eroglu@ege.edu.tr

**Keywords:** *Astragalus flavescens*, legumes, nodulation genes, *Phyllobacterium*, rhizobia

## Abstract

A novel strain of Gram-negative, rod-shaped aerobic bacteria, identified as IY22, was isolated from the root nodules of *Astragalus flavescens*. The analysis of the 16S rDNA and *recA* (recombinase A) gene sequences indicated that the strain belongs to the genus *Phyllobacterium*. During the phylogenetic analysis, it was found that strain IY22 is closely related to *P. trifolii* strain PETP02^T^ and *P. bourgognense* strain STM 201^T^. The genome of IY22 was determined to be 6,010,116 base pairs long with a DNA G+C ratio of 56.37 mol%. The average nucleotide identity (ANI) values showed a range from 91.7% to 93.6% when compared to its close relatives. Moreover, IY22 and related strains had digital DNA-DNA hybridization (dDDH) values ranging from 16.9% to 54.70%. Multiple genes (including *nodACDSNZ*, *nifH/frxC*, *nifUS*, *fixABCJ*, and *sufABCDES*) associated with symbiotic nitrogen fixation have been detected in strain IY22. Furthermore, this strain features genes that contribute to improving plant growth in various demanding environments. This study reports the first evidence of an association between *A. flavescens* and a rhizobial species. Native high-altitude legumes are a potential source of new rhizobia, and we believe that they act as a form of insurance for biodiversity against the threats of desertification and drought.

## 1. Introduction

The genus *Phyllobacterium* is a member of the order Rhizobiales and includes Gram-negative, motile, non-spore-forming, aerobic, and chemoorganotrophic species [1]. While Zimmermann (1902) reported the first isolation of strains from the *Phyllobacterium* genus, it was Knösel (1962) who first used the name *Phyllobacterium* for bacteria growing in leaf nodules. Many bacteria within this group are associated with plants and can also be found in various habitats such as water, soil, rhizosphere, and root nodules, and in association with single-celled organisms [2,3,4,5,6]. To the present day, sixteen species of *Phyllobacterium* have been documented in the source provided (https://lpsn.dsmz.de/search?word=Phyllobacterium (accessed on 30 December 2023)) (Table 1).

Rhizobia colonization is a crucial factor that affects the productivity of leguminous plants. These bacteria inhabit symbiotic nodules formed in the roots of leguminous plants [18]. The relationship between the bacteria and the legume in this symbiotic life is highly specific. Specific signal exchanges occur between legumes and the bacteria forming these nodules. Rhizobia synthesize lipochitin oligosaccharides known as Nod factors during these interactions [19]. Rhizobial species trigger the development of root nodules by activating Nod factors. This symbiotic relationship streamlines the fixation of atmospheric dinitrogen within these nodules, thereby aiding rhizobia in generating the necessary nitrogen source for the plant [20]. 

Legumes and rhizobia have a mutually beneficial relationship that positively impacts the ecosystem in various ways. This includes mechanisms for metal tolerance/resistance, plant growth promotion, nitrogen fixation, phytohormone production, and phosphorus solubility. PGPR (plant growth-promoting rhizobacteria) exert their active principles through enzymes and secondary metabolites [21]. Recently developed genome-based approaches and bioinformatic tools offer the opportunity to examine the entire genome of rhizobial strains. This enables the investigation of genes that encode beneficial enzymes, which facilitate symbiotic interactions and modulate plant hormone levels, as well as biosynthetic gene clusters that encode active compounds such as antibiotics [22,23,24]. Genome-wide analyses enable the unveiling of the entire arsenal of a PGPR strain, thereby revealing the full potential of the strain for future field applications. This allows the characterization of bacterial strains as potential biofertilizer candidates, considering even possible virulence factors for security purposes [25,26].

With the advancement of genomic techniques, the use of bioinformatic approaches based on whole-genome sequence analysis is now recommended as minimal standards for the identification of new prokaryotic taxa, in addition to classical DNA hybridization techniques. Adequate quantitative comparisons should be conducted between the genomes of new species and the type strains of closely related species. Currently, the most common criteria for assessing genomic similarities between strains are obtained through whole-genome average nucleotide identity (ANI) and digital DNA–DNA hybridization (dDDH) techniques [27]. In rhizobia taxonomy, the primary approach also relies on genomic sequences, ANI, and dDDH values among strains [28,29].

In this study, we isolated rhizobial strain IY22 from the root nodules of *Astragalus flavescens* Boiss., an endemic and rare species found in Bozdağ, Türkiye. The 16S rDNA sequence of strain IY22 exhibited close similarity to *P. trifolii* strain PETP02^T^, whereas the *recA* sequence displayed significant homology with *P. bourgognense* strain STM 201^T^. The analysis of complete genome sequences, dDDH, ANI, and biochemical assessments revealed that strain IY22 is a new species within the genus.

## 2. Materials and Methods

### 2.1. Sample Collection and Isolation 

*Astragalus flavescens* was collected at an altitude of 1450 m in Ödemiş, Bozdağ (Izmir, Türkiye) at coordinates 38°21′35.94″ N and 28°07′05.90″ E (Figure 1A). A research license to investigate endemic flora was acquired from the Ministry of Agriculture and Forestry’s General Directorate of Agricultural Research and Policies. The study was conducted in compliance with applicable institutional, national, and international protocols and legislation. Plant roots were collected from soil excavated to a depth of approximately 40–50 cm using a hand hoe, and placed in a sample bag containing moistened cotton to prevent the drying of the nodules. Within 2 h, the samples were transported to the Laboratory of Basic and Industrial Microbiology at Ege University. In addition, the plant specimen was identified at the Botanical Garden and Herbarium Research Application Centre of Ege University and registered under the accession number EGE-43727 in the Ege University Herbarium collection.

The nodules were meticulously detached from the plant’s root system and rinsed thoroughly with sterile distilled water to eliminate soil particles (Figure 1B). The nodule surface was sterilized by immersing it in 70% ethanol for 3 min, followed by a 2 min treatment with a mercury (II) chloride solution (HgCl_2_). Following the procedure, the nodules were washed four times with sterile distilled water. Subsequently, a nodule was sectioned in paraffin and examined under a microscope. At the same time, the nodules were crushed using a sterile glass rod. The homogenized nodule tissue was inoculated onto Yeast Extract Mannitol Agar (YEMA) using the spread plate technique [30]. Petri dishes were incubated at 28 °C for 3–10 days. Bacterial cultures that were to be used for subsequent phenotypic and molecular analysis were purified from a solitary colony after 3–4 days of incubating on YEMA, at a temperature of 28 °C.

### 2.2. Phylogeny of 16S rDNA and recA Genes

Genomic DNA was prepared following the High Pure PCR Template Preparation Kit protocol (Roche Applied Science, Mannheim, Germany). The DNA concentration was controlled by the Qubit™ 4 (Invitrogen, New York, NY, USA). The 16S ribosomal DNA (rDNA) and DNA recombinase A (*recA*) gene sequences of strain IY22 were amplified with PCR using the 27F-1492R and recAF-recAR primer pairs, respectively (Appendix A). The FastStartTM Taq DNA Polymerase was used for all PCR amplifications (Roche Applied Science, Mannheim, Germany). The PCR products were sequenced at Letgen Biotechnology Company (Izmir, Türkiye).

Both obtained and related sequences were retrieved from the NCBI (National Center for Biotechnology Information) GenBank databases using the BLASTn search program. Phylogenetic trees were configured separately for the 16S rRNA and *recA* genes according to the Neighbor joining (NJ) method integrated in Molecular Evolutionary Genetics Analysis (MEGA) software version 11 [31].

### 2.3. Phenotypic Characterization

Strain IY22 obtained from the root nodule of *A. flavescens* was characterized. The characterization included Gram test, catalase and oxidase test, and macroscopic and microscopic observations (Olympus CX21,Tokyo, Japan). The growth on YEMA containing Congo red dye (YEMA + CR) was also observed. The ability to grow at different temperatures (22, 28, 37, and 39 °C) in the presence of different NaCl concentrations (1, 2, and 3%) and at different pH (5, 6, 7, and 8) was determined on the YM Broth medium. Other physiological and biochemical tests were also carried out using API (20E, 20NE) test kits (bioMérieux, Marcy l’Etoile, France). API tests were performed according to the manufacturer’s instructions.

### 2.4. Genome Analysis 

The genomic DNA (50 ng/μL) of strain IY22 was sequenced using the Illumina NovaSeq 6000 sequencing platform with 150 × 2 bp reads with a ~340 bp insert size library by Refgen Biotechnology Inc. (Ankara, Türkiye). Around 2.2 gigabases (Gb) of data were generated, consisting of 7,623,320 reads, using the Illumina NovaSeq 6000 sequencing platform (San Diego, CA, USA). The de novo assembly was carried out using Shovill v1.0.4. with the default settings. Genome annotation was performed using the NCBI Prokaryotic Genome Annotation Pipeline via the NCBI Genome Submission Portal. BioProject ID: PRJNA923719, GenBank Accession number: JAQMHX000000000.1. 

The ANI values for strain IY22 and its reference bacteria were computed using the online OrthoANIu algorithm (OrthoANI employing USEARCH) [32]. To calculate the dDDH values, Formula 2 was applied through the server-based Genome-to-Genome Distance Calculator, version 3.0 (http://ggdc.dsmz.de/distcalc2.php (accessed on 20 September 2023)). Taxonomy was determined using the Type (Strain) Genome Server (TGYS) (http://tygs.dsmz.de/ (accessed on 20 September 2023)) with default parameters [33]. Genome sequences of closely related species were acquired from the EzBioCloud and NCBI databases.

Gene functions were determined utilizing the Rapid Annotation Subsystem Technology (RAST) (https://rast.nmpdr.org/ (accessed on 5 February 2023)) and the Pathosystems Resource Integration Center (PATRIC) (https://www.bv-brc.org/ (accessed on 5 February 2023)) web-based pipelines [34,35]. Also on the SEED website, the SEED viewer method was used to perform a functional genomic analysis. Furthermore, the prediction of CRISPR repeats was conducted through the utilization of the CRISPRfinder web server (https://crisprcas.i2bc.paris-saclay.fr (accessed on 10 February 2023)) [36]. Prophages were detected using the Prophage Hunter (https://pro-hunter.genomics.cn (accessed on 10 February 2023)) [37].

## 3. Results and Discussion

The Sanger sequencing of the 16S rRNA gene sequence of strain IY22 (accession number OQ301811) showed 98.8% similarity to *P. trifolii* strain PETP02^T^ after BLASTn search of the NCBI nucleotide database. The *recA* sequence analysis resulted in a close relationship with *P. bourgognense* strain STM 201^T^ with a similarity of 96.02%. Our findings strongly support the idea that strain IY22, which we investigated in this study, forms a monophyletic group with previously known *Phyllobacterium* species, indicating its membership of the *Phyllobacterium* genus (Figure 2 and Figure 3). However, differences in clusters and topology suggest that a phylogenetic analysis based on individual genes may be inconsistent due to their unique evolutionary histories. These paradoxes highlight that constructing an accurate phylogenetic relationship among rhizobial species solely based on individual genes is unlikely [38,39].

Strain IY22 formed a faint pink colony on YEMA medium containing Congo red and formed small colonies (<2 mm in diameter). The cells are Gram-negative, catalase-, and oxidase-positive. Microscopic observations showed that the strain was rod-shaped bacteria (Figure 1C,D). The pH range for growth is pH 5–8 (optimal growth occurs at pH 7). Growth occurs in the presence of 0–2% (*w/v*) NaCl. The optimal growth temperature is 28 °C. Other physiological and biochemical tests were carried out using API systems following the manufacturer’s instructions, and the results were read after 3 days of incubation at 28 °C. Based on the findings, strain IY22 exhibited differences from the *P. trifolii* PETP02^T^ and *P. bourgognense* STM 201^T^ strains in terms of its ability to produce acids from sucrose, trehalose, rhamnose, and raffinose, as well as its indole production and citrate assimilation (Table 2). Specifically, IY22 displayed positive acid production from sucrose, trehalose, and raffinose, while PETP02^T^ exhibited negative results. Furthermore, IY22 showed positive indole production, in contrast to STM 201^T^ which showed a negative result. Additional phenotypic traits of strain IY22 are elaborated upon in the subsequent species description.

The complete genome size of strain IY22 was 6,010,116 bp. The DNA G+C content of strain IY22 was 56.37 mol% (Table 3). In the close relatives of strain IY22, *P. trifolii* PETP02^T^ and *P. bourgognense* STM 201^T^, the G+C contents are 56.4 mol% and 54 mol% (Tm), respectively [9,10]. The ANI values for strain IY22 and its closely related *Phyllobacterium* species range from 91.7 to 93.6. The dDDH values, calculated using the suggested Formula 2, exhibit a range of 16.9% to 54.70% for the IY22 strain and the closely related *Phyllobacterium* strains. In terms of the dDDH value, strain IY22’s highest relationship was observed with *P. trifolii* PETP02^T^ (54.70%). Nevertheless, these values, which are typically utilized as thresholds for bacterial species identification, notably remain below the commonly accepted thresholds of 95–96% (ANI) and 70% (DDH) [40,41]. A phylogenetic tree was generated by analyzing genomes from the nearest relatives of strain IY22 using the TGYS platform (Figure 4).

The genome of strain IY22 consists of 65 contigs, comprising one chromosome and no plasmids. The chromosome contains 3 identical rRNA operons and 46 tRNA genes. Of the 6470 predicted CDS, 2534 were coding hypothetical proteins, 3936 were proteins with functional assignments, 1232 were proteins with EC number assignments, 4252 were proteins with PATRIC genus-specific family (PLfam) assignments, and 4701 were proteins with PATRIC cross-genus family (PGfam) assignments. One CRISPR locus was found in the genome via the CRISPRCasFinder (https://crisprcas.i2bc.paris-saclay.fr (accessed on 10 February 2023)). Six prophages were detected using the Prophage Hunter (https://pro-hunter.genomics.cn (accessed on 10 February 2023)). Genomic and protein characteristics are condensed in Table 3 and Table 4. The genome of strain IY22 was additionally mapped to the seed subsystem to attain high-quality genome annotation via RAST. The distributions of genes linked to subsystems in 27 different categories are shown in Figure 5.

The analysis of carbohydrate metabolism in *Phyllobacterium* genus species was compared to GenBank reference genomes, revealing similar profiles. In the IY22 genome, 215 genes associated with pathways such as xylose utilization, chitin and N-acetylglucosamine metabolism, maltose and maltodextrin utilization, trehalose biosynthesis, lactate utilization, one-carbon metabolism by tetrahydropterines, and various others were identified (Appendix A). The genomic analysis of strain IY22 identified *fdoGH*, which encodes formate dehydrogenase, a key player in C1 metabolism. Formate metabolism is crucial in Rhizobiales for nitrogen fixation and formate-dependent respiration [42]. Additionally, IY22 also possesses genes for selenocysteine biosynthesis, which were first identified in *S. meliloti* [43].

In recent times, genomic approaches have been suggested as an alternative to traditional chemotaxonomy for identifying bacterial species [44]. An in silico chemotaxonomy methodology was initially applied to *Corynebacterium* and *Turicella,* and subsequently to *Aliamphritea* species [45,46]. In this study, we used genome sequences to predict chemotaxonomic characteristics of strain IY22. A comparative genomic analysis of closely related *Phyllobacterium* species revealed that strain IY22 possesses the essential type II fatty acid biosynthesis (FASII) pathway. The fatty acid metabolism pathway based on KEGG is illustrated in Figure 6. This pathway is responsible for synthesizing three major fatty acids: C16:0, C16:1, and C18:1 [47]. The genome of strain IY22 encodes fabA and fabB, indicating the potential to produce C16:1ω7c and C18:1ω7c. The enzymes responsible for the FASII pathway in the IY22 genome were identified and are listed in Appendix A. All genes encoding these enzymes are present in *P. trifolii* PETP02^T^. The predominant cellular fatty acids produced through these enzymes, crucial to the FASII pathway determinants, including myristic acid (14:0), palmitic acid (16:0), 3-Hydroxypentadecanoic acid (15:0 3-OH), margaric acid (17:0), 3-Hydroxyhexadecanoic acid (16:0 3-OH), stearic acid (18:0), cis-vaccenic acid (C18:1ω7c), cis-9,10-Methyleneoctadecanoic acid (19:0 cyclo ω8c), oleic acid (18:1), eicosadienoic acid (C20:2ω6,9c), and arachidic acid (20:0) [9]. 

Certain fatty acids, especially polyunsaturated ones, can be synthesized through various enzymatic pathways. For example, stearic acid (18:0) is typically desaturated to produce C18:1ω7c. This desaturation process is catalyzed by an enzyme called stearoyl-ACP desaturase, which converts the methylene group of stearic acid into a double bond, resulting in the production of oleic acid (18:1) and subsequently cis-vaccenic acid (C18:1ω7c) [48]. C20:2ω6,9c is a polyunsaturated fatty acid (PUFA) typically synthesized through the desaturation of linoleic acid (C18:2ω6c). The desaturation of linoleic acid is catalyzed by one or more desaturase enzymes, resulting in the production of oleic acid (C18:1) and subsequently eicosadienoic acid (C20:2ω6,9c) [49]. These metabolic pathways, particularly those involving specific enzymatic steps such as fatty acid desaturation, guide the synthesis of different types of fatty acids. The amino acid sequences of fatty acid desaturases encoded in the genomes of strain IY22 and closely related species were aligned, revealing diversity in conserved histidine clusters (Figure 7). Changes in the active regions of such enzymes can lead to differences in reaction steps, contributing to the uniqueness and diversity of organisms’ lipid metabolism.

The genome of strain IY22 contains all the necessary genes for producing phosphatidylglycerol (PG) and phosphatidylethanolamine (PE), including *plsX*, *plsY*, *plsC*, *cdsA*, *pssA*, *psd*, *pgsA*, and *pgpA*. It also has *clsA*, which is responsible for diphosphatidylglycerol (DPG) production. Additionally, the *clsII* gene with eukaryotic-like characteristics was identified in the genome (Appendix A). Strain IY22 is capable of utilizing two different pathways for cardiolipin synthesis, similar to the mechanisms observed in *Streptomyces coelicolor* or *Agrobacterium tumefaciens* [50].

The predominant respiratory quinone in previously described *Phyllobacterium* species is ubiquinone-10 [9,10,11,12,16]. The comparative analysis of the genome of strain IY22 revealed the presence of the *ubiABEGX* and *pqqC* genes. The *ubiE* gene catalyzes the conversion of demethylmenaquinone to menaquinone (Appendix A). These results indicate that, similar to other members of the genus, menaquinone-10 is the predominant quinone in strain IY22.

The genome of strain IY22 contains the crucial *LpxABCDH* genes necessary for lipid A biosynthesis (Appendix A). Notably, these genes exhibit a remarkable level of conservation among rhizobial species, encompassing *Mesorhizobium*, *Bradyrhizobium*, and *Phyllobacterium* [51].

This study presents the initial comparison of in silico chemotaxonomic data for a rhizobial species. As the number of genome sequences in databases increases, in silico chemotaxonomy is expected to become applicable to numerous bacterial species in the future. Additionally, this approach may be effective in predicting chemotaxonomic characteristics of unculturable species using metagenomic data. However, there is a need for further advancements in predicting the structure of enzymes responsible for fatty acid metabolism and their biochemical kinetics. This is crucial because current methods are not well suited to accurately determine the quantitative amounts of predicted fatty acids [46].

A general screening was carried out for genes essential for promoting plant growth under different stresses. Several genes related to iron uptake (17), nitrogen metabolism (8), virulence, disease and defence (32), membrane transport (54), stress response (50), and sulfur metabolism (18) were found in strain IY22 (Appendix A). The variety of habitats indicates that *Phyllobacterium* has acquired an impressive aptitude for adapting to its surroundings. Moreover, their non-pathogenic attributes and capacity to ‘interact’ with plant tissues render them intriguing subjects for research aimed at exploring their potential to promote plant growth [52,53].

In accordance with their previously proposed functions, several genes associated with nodule development (*nodACDSNZ*) in strain IY22 were identified (Table 5). The β-1,4-linked D-glucosamine residues form the Nod factor backbone, which is N-acylated on the non-reducing end and O-acetylated on the other residues. *nodC* encodes N-acetylglucosaminyl transferase [54]. The expression of nod genes is controlled by flavonoid-activated NodD proteins, which act as LysR-type transcriptional regulators [55]. NodS is a S-adenosylmethionine-dependent methyltransferase that methylates chitooligosaccharides that are deacetylated at the non-reducing end. NodZ is responsible for the fucosylation of nodulation signals [56,57]. Functionally conserved nod genes are commonly used as target genes in studies of rhizobial diversity [58]. In this study, genes encoding nod factors were identified in the genome of IY22 and a phylogenetic tree was constructed by comparing *nodC* amino acid sequences, as shown in Figure 8. However, the *nodC* phylogeny of IY22 displays incongruity with the 16S rRNA taxonomy. Classifying rhizobia according to their symbiotic properties is a major challenge due to the complexity of the plant molecular mechanisms that determine host specificity [59]. This classification is thus largely reliant on the growth conditions of the leguminous plant.

In conjunction with the nod genes, the IY22 genome also harbors key genes responsible for regulating nitrogen fixation, including *nifH/frxC*, *nifUS*, *sufABCDES*, *NtrBCYX*, *fixABCJ*, *GlnK*, *glnD*, and *cysE* (Table 5). Biological nitrogen fixation is driven by molybdenum-dependent nitrogenase, a sophisticated enzyme consisting of a two-component protein (MoFe protein and Fe protein). The *nifHDK* genes encode structural subunits, while the *nifBENQVYX* genes encode the synthesis and attachment of FeMo-co to nitrogenase [60]. The assembly of the Fe-S cluster is managed by NifU and NifS [61]. The formation and integration of nitrogenase metalloclusters necessitate the production of initial Fe-S units, which can be sourced from either general or specialized biosynthetic systems [62]. Iron–sulfur clusters serve as essential metal cofactors vital for critical biological processes, including DNA repair, respiration, nitrogen fixation, and photosynthesis [63]. The mechanisms underlying Fe-S cluster biosynthesis in various bacterial species involved in nitrogenase are still not well understood. In IY22, a fully functional nif operon was not observed; thus, it is likely that the NifS and SufCDB proteins play the role of a specific sulfur donor and molecular scaffold, sequentially, in the assembly of nitrogenase Fe-S clusters, as observed in *P. polymyxa* WLY78 [64]. In contrast, the *fixABCX* operon in IY22 may be associated with nitrogen fixation, as in *Azospirillum* [65]. Moreover, the genetic constituents and configurations of genes responsible for nitrogen fixation can exhibit significant diversity across various diazotrophic organisms [66]. 

## 4. Conclusions

The exploration of legume–plant interactions can provide invaluable insights into the realms of evolutionary biology and fundamental ecological processes. The studies conducted in this regard can act as a form of insurance for biodiversity against the threats of desertification and drought. Moreover, in the context of livestock farming, it is well established that plants cultivated at high altitudes exhibit elevated nutritional values. Additionally, it can be hypothesized that unexplored attributes might exist within plants and microorganisms that have adapted to climatic conditions prevalent at high altitudes, such as low temperatures, altered atmospheric pressure, and heightened UV radiation. Conducting a more comprehensive investigation into the microbial diversity intertwined with high-altitude flora holds paramount importance. Such an endeavor will facilitate the identification of novel bacterial taxa and bacterial–plant symbiotic relationships with the potential to offer valuable contributions to agricultural domains.

Chemical fertilizers have detrimental effects on agricultural land ecosystems. To mitigate these negative impacts, alternative methods such as biological nitrogen fixation are increasingly important. Research on prokaryotic taxa associated with legumes has been intense. Nevertheless, a rhizobial species has not yet been identified in *A. flavescens*. This study presents the initial findings of an association between *A. flavescens* and a rhizobial species. This belated discovery may be attributed to the fact that past research has primarily examined diversity at lower altitudes. Our investigation indicates that the indigenous vegetation at high altitudes provides a probable source for the identification of novel rhizobial species.

### Description of “Candidatus Phyllobacterium onerii” sp. nov.

*Phyllobacterium onerii*: o-ne-ri-i N.L. gen. n. onerii named after Mehmet Öner, an important scientist in the field of microbiology, founder of the Basic and Industrial Microbiology Department of Ege University.

The colonies that develop aerobically in YEMA at 28 °C are small, scaly, and white. The cultures grown appear to have a faint pink color when grown in YEMA containing Congo Red. The cells are rod-shaped, Gram-negative, catalase-, and oxidase-positive. The cells can grow in the presence of NaCl concentrations up to 2% (*w*/*v*). The temperature range for growth is 4–37 °C; however, the optimal growth temperature is 28 °C. The pH range for growth is 6–8 (optimal growth occurs at pH 7). Arginine dehydrolase, β-galactosidase, and gelatinase were negative. The hydrolysis of aesculin was weak. Lysine decarboxylase, ornithine decarboxylase, tryptophan deaminase, and urease were positive. Indole production and citrate assimilation were positive. Melibiose and amygdalin fermentation were negative. Glucose, L-arabinose, mannose, inositol, sorbitol, rhamnose, sucrose, mannitol, N-acetylglucosamine, maltose, and malate are carbon sources. Gentiobiose is weakly used. It does not grow on caproate, adipate, or phenylacetate. The in silico genome analysis indicates that strain IY22 possesses the FASII biosynthesis pathway, enabling it to produce myristic acid (14:0), palmitic acid (16:0), 3-hydroxypentadecanoic acid (15:0 3-OH), margaric acid (17:0), 3-Hydroxyhexadecanoic acid (16:0 3-OH), stearic acid (18:0), cis-vaccenic acid (C18:1ω7c), cis-9,10-methyleneoctadecanoic acid (19:0 cyclo ω8c), oleic acid (18:1), eicosadienoic acid (C20:2ω6,9c), and arachidic acid (20:0). Strain IY22 encodes a comprehensive set of genes essential for the synthesis of phosphatidylglycerol (PG) and phosphatidylethanolamine (PE), including *plsX*, *plsY*, *plsC*, *cdsA*, *pssA*, *psd*, *pgsA*, and *pgpA*. Additionally, it includes *clsA*, which is responsible for the production of diphosphatidylglycerol (DPG).

Strain IY22 was isolated from the root nodule of the İzmir/Bozdağ endemic legume plant *Astragalus flavescens*. The DNA G+C content is 56.37 mol%. BioProject ID: PRJNA923719, GenBank Accession number: JAQMHX000000000.1. Genome assembly: GCA_028331445.1.

## Figures and Tables

**Figure 1 microorganisms-12-00336-f001:**
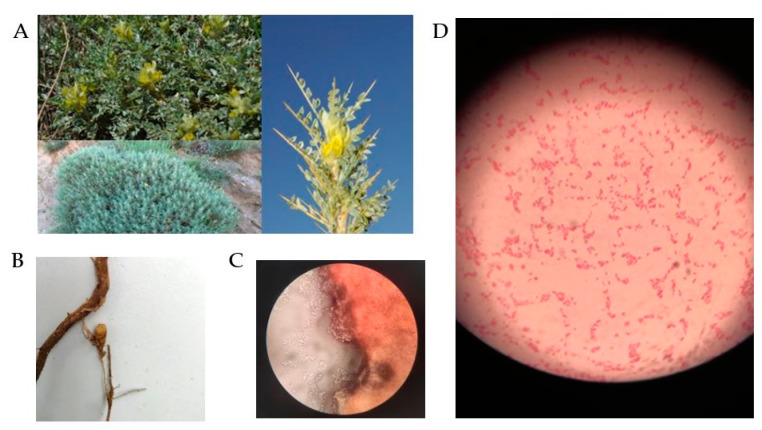
*Astragalus flavescens*, an endemic legume found in Bozdağ (**A**), is shown with its symbiotic nodule (**B**). Bacillus-shaped bacteria were observed on microscopic analysis of the nodule section (**C**). Gram-negative stained bacilli, under 100× magnification (**D**).

**Figure 2 microorganisms-12-00336-f002:**
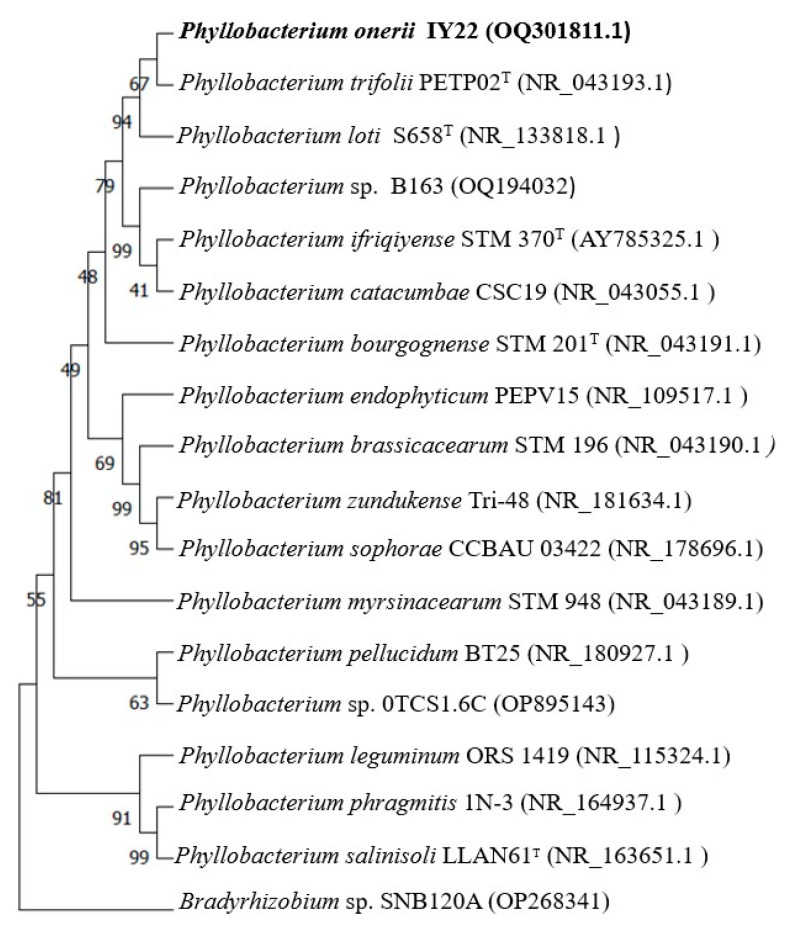
Neighbor-joining 16S rDNA phylogenetic tree of *Phyllobacterium onerii* sp. nov IY22 and closely related species. The stability of the groupings was estimated using bootstrap analysis with 1000 pseudoreplications. *Bradyrhizobium* sp. was added as an outgroup.

**Figure 3 microorganisms-12-00336-f003:**
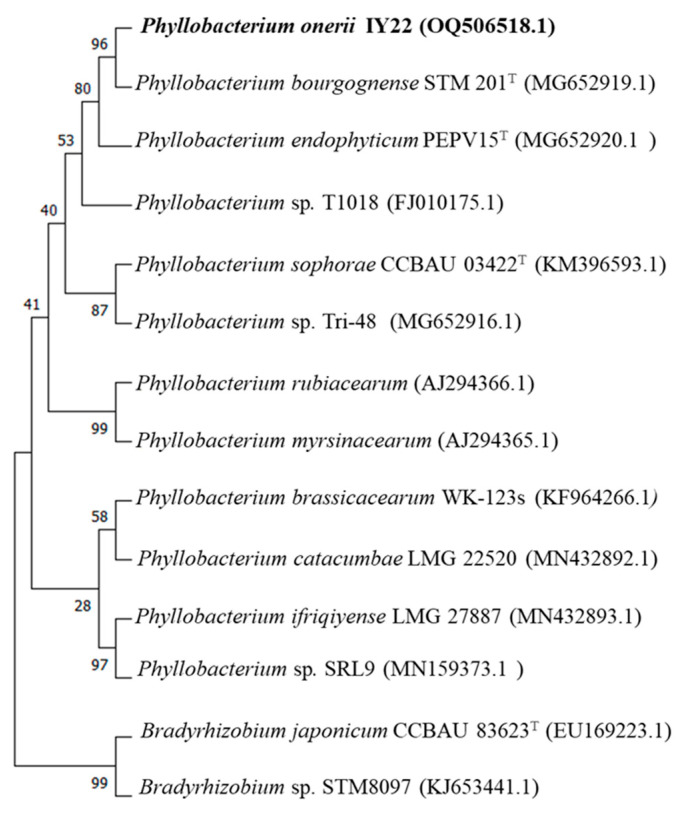
Phylogenetic tree based on *recA* gene sequences showing relationships between strain IY22 and strains of related species. The tree was constructed using the neighbor-joining method.

**Figure 4 microorganisms-12-00336-f004:**
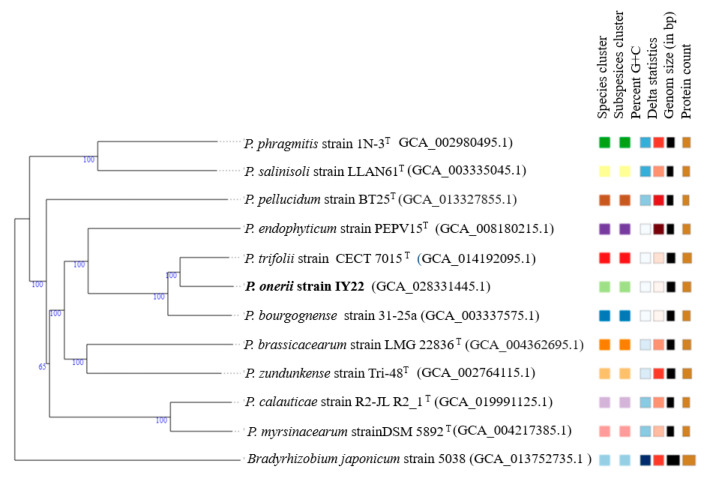
The tree was built with FastME v2.1.6.1 using whole genome-based GBDP distances from the Type Strain Genome Server (TYGS). Branch lengths are scaled with the GBDP distance formula D5. The genome size varies between 4,660,625 and 9,226,248 bp. The protein number ranges from 4558 to 8676. The percent G+C ratios range from 56.37 to 63.66.

**Figure 5 microorganisms-12-00336-f005:**
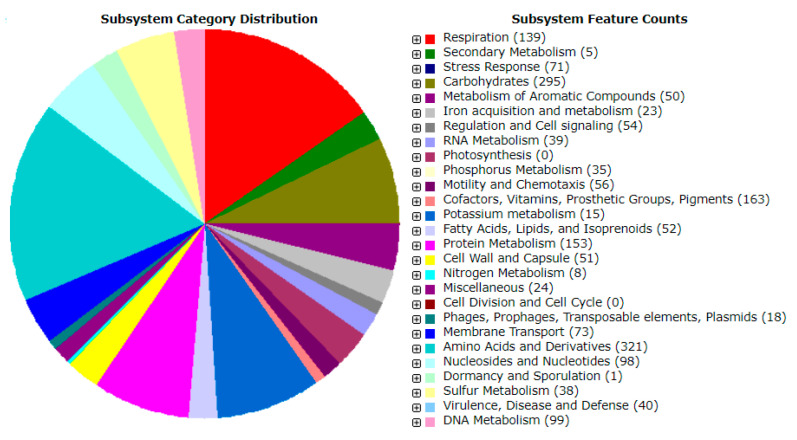
Genes related to subsystems and their distribution in different categories.

**Figure 6 microorganisms-12-00336-f006:**
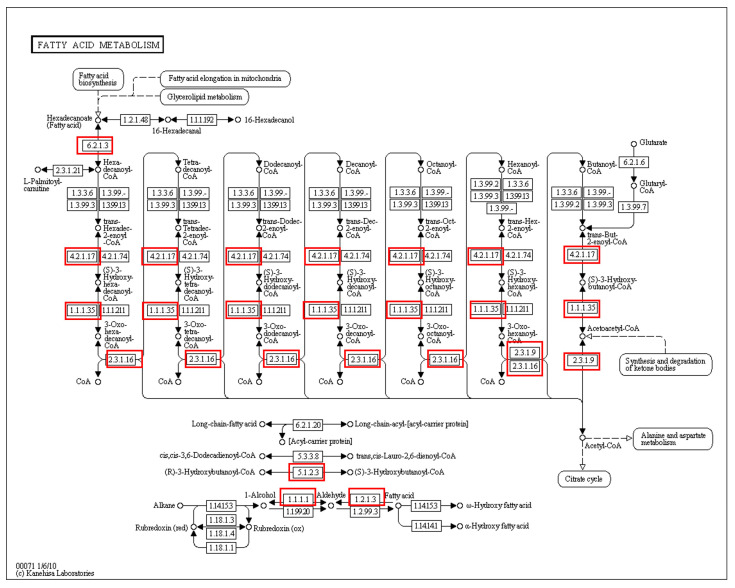
The fatty acid metabolism is illustrated using a KEGG pathway diagram. The proteins highlighted in red boxes were identified in strain IY22. Numbers indicate E.C. numbers of enzymes.

**Figure 7 microorganisms-12-00336-f007:**
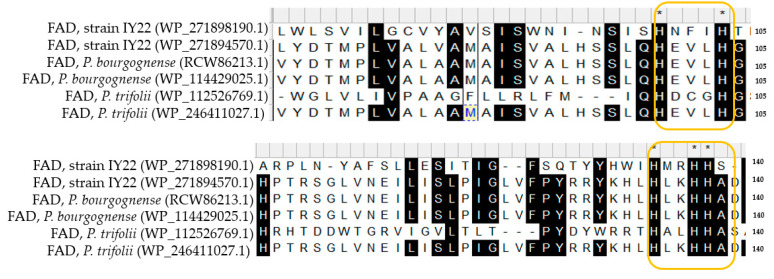
A multiple alignment of amino acid sequences of fatty acid desaturases (FAD). The conserved residues ‘HXXXH’ and ‘HXXHH’ are highlighted in yellow boxes. Conserved amino acid motifs are highlighted in black. The asterisk (*) indicates residues that are critical for enzyme function.

**Figure 8 microorganisms-12-00336-f008:**
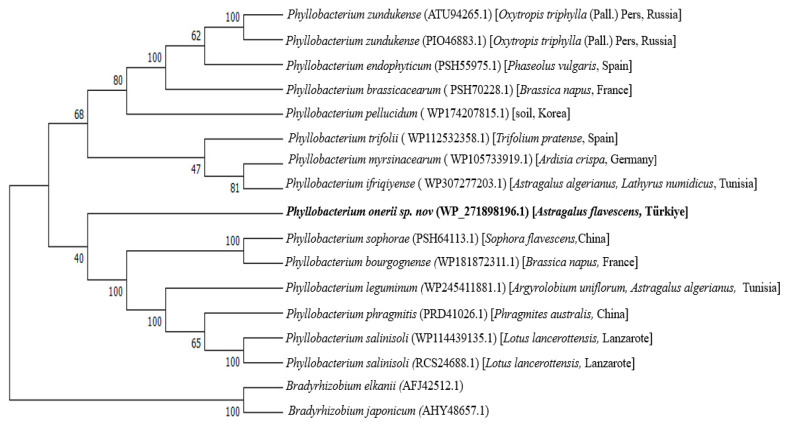
Phylogenetic tree based on *nodC* amino acid sequences showing relationships between strain IY22 and strains of related species. The tree was constructed using the neighbor-joining method.

**Table 1 microorganisms-12-00336-t001:** List of *Phyllobacterium* species and sources.

Species	Source	Reference
*P. myrsinacearum*	*Ardisia crispa*, Germany	[7]
*P. rubiacearum (junior subjective synonym*, *P. myrsinacearum)*	*Pavetta* leaf nodules, Belgium	[8]
*P. trifolii*	*Trifolium pratense*, Spain	[9]
*P. catacumbae*	Roman catacombs of Saint Callixtus, Italy	[4]
*P. bourgognense*	*Brassica napus*, France	[1]
*P. brassicacearum*	*Brassica napus*, France	[1]
*P. ifriqiyense*	*Astragalus algerianus* and *Lathyrus numidicus*, Tunisia	[1]
*P. leguminum*	*Argyrolobium uniflorum* and *Astragalus algerianus*, Tunisia	[1]
*P. endophyticum*	*Phaseolus vulgaris*, Spain	[10]
*P. loti*	*Lotus corniculatus*, Uruguay	[11]
*P. sophorae*	*Sophora flavescens*, China	[12]
*P. salinisoli*	*Lotus lancerottensis*, Lanzarote	[13]
*P. zundukense*	*Oxytropis triphylla* (Pall.) Pers, Russia	[14]
*P. phragmitis*	*Phragmites australis*, China	[15]
*P. pellucidum*	soil, Korea	[16]
*P. calauticae*	sediment, Denmark	[17]

**Table 2 microorganisms-12-00336-t002:** Differential phenotypic characteristics of *Phyllobacterium onerii* sp. Nov. IY22 and its close relatives *P. trifolii PETP02^T^* and *P. bourgognense* STM 201^T^. +, positive; −, negative; w, weak; nd, no data.

Characteristic	*P. onerii* IY22 (This Study)	*P. trifolii* PETP02^T^ [9]	*P. bourgognense* STM201^T^ [1]
Sucrose	+	−	nd
Trehalose	+	−	w
Rhamnose	+	w	+
Raffinose	+	−	+
Citrate assimilation	+	−	+
Indole production	+	−	−

**Table 3 microorganisms-12-00336-t003:** Annotated genome features of *P. onerii* and close relatives.

Attribute	*P. onerii * (GCA_028331445.1)	*P. trifolii* (GCA_014192095.1)	*P. bourgognense* (GCA_003337575.1)
Contigs	65	69	61
GC Content	56.37	56.4	56.5
Plasmids	0	0	0
Contig L50	5	9	7
Genome Length	6,010,116 bp	6,306,857 bp	5,618,596
Contig N50	438,728	236,202	228,269
Chromosomes	1	1	1
CDS	6470	6715	5995
tRNA	46	41	46
rRNA	3	3	3

**Table 4 microorganisms-12-00336-t004:** Protein features of strain IY22.

Protein Features	Number
Hypothetical proteins	2534
Proteins with functional assignments	3936
Proteins with EC number assignments	1232
Proteins with GO assignments	1087
Proteins with pathway assignments	952
Proteins with PATRIC genus-specific family (PLfam) assignments	4252
Proteins with PATRIC cross-genus family (PGfam) assignments	4701

**Table 5 microorganisms-12-00336-t005:** Genes associated with symbiotic nitrogen fixation in strain IY22. Identical groups of proteins are indicated by the asterisk (*) symbol.

Genes	Function of Gene Product or Identity	Length (aa)	Acc. Number
*nodA*	N-acyltransferase	351	WP_271896533
*nodC*	N-acetylglucosaminyltransferase	423	WP_271898196.1
*nodD*	LysR family transcriptional regulator	298	WP_271898124.1
*nodS*	SAM-dependent methyltransferase	281	WP_271898982.1
*nodN*	Nodulation protein N (MaoC family dehydratase)	159	WP_271898282.1
*nodZ*	GDP-L-fucose synthetase	333	WP_271892710.1
*nifU*	Fe-S cluster biogenesis protein	192	WP_114431505.1 NZ_JAQMHX010000004.1 217482-218060
*nifH/frxC*	4Fe-4S binding protein	666	WP_271899276.1
*nifS*	Cysteine desulfurase	391	WP_271893808.1
*sufA*	Fe-S cluster assembly protein	124	WP_271893816.1
*sufB **	Fe-S cluster assembly protein	509	WP_112530944.1 NZ_JAQMHX010000001.1 1104702-1106231
*sufC **	Fe-S cluster assembly ATPase	251	WP_271893810.1 NZ_JAQMHX010000001.1 1106527-1107282
*sufD*	Fe-S cluster assembly protein	423	WP_271893811.1
*sufE **	Sulfur acceptor protein	146	WP_224505514.1 NZ_JAQMHX010000001.1 576195-576635
*sufS*	Cysteine desulfurase	413	WP_271893813.1
*NtrB*	Nitrogen metabolism regulation gene	384	WP_271896598.1
*NtrC **	Nitrogen metabolism regulation gene	486	WP_271896598.1
*NtrX **	Nitrogen metabolism regulation gene	486	WP_224506676.1 NZ_JAQMHX010000005.1 410594-412054 (+)
*GlnK **	Nitrogen regulatory protein P-II	112	WP_008123000 NZ_JAQMHX010000012.1 108434-108772 (+)
*glnD*	[Protein-PII] uridylyltransferase	937	WP_271895489.1
*cysE*	Serine acetyltransferase	274	WP_271896446.1
*fixA*	Electron transfer flavoprotein beta subunit	250	WP_271897294.1
*fixB*	Electron transfer flavoprotein alfa subunit	309	WP_271897292.1
*fixC*	Flavoprotein-ubiquinone oxidoreductase	555	WP_271893276.1
*fixj*	Response regulator low-oxygen conditions	124	WP_271898449.1

## Data Availability

All sequence data support the findings of this study have been deposited in GenBank (https://www.ncbi.nlm.nih.gov/genbank/ (accessed on 2 February 2023)) with accession numbers OQ301811-OQ506518 and JAQMHX000000000.1. The complete genome data has been assigned BioProject ID PRJNA923719 and is now accessible to the public.

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
