# Peer review of "Genomic Insights into the Symbiotic and Plant Growth-Promoting Traits of “Candidatus Phyllobacterium onerii” sp. nov. Isolated from Endemic Astragalus flavescens"

_microorganisms, 2024, doi:10.3390/microorganisms12020336_

Round 1

Reviewer 1 Report

Comments and Suggestions for Authors

Dear authors,

The manuscript microorganisms-2816636 entitle “Genomic insights into the symbiotic and plant growth-promoting traits of Phyllobacterium onerii sp. nov isolated from endemic Astragalus flavescens”. The manuscript shows the phenotypic and some genomics characteristics, however, before that was accept it the authors needs considered the following comments.

Specific comments

1.     The authors need provide a carbon source assimilation by Phyllobacterium onerii and verified some metabolic pathway with an in silico analysis.

Major points

·       The introduction paragraph needs to be improved.

·       In the phylogenetic tree the species names must be written in italics and the accession number must be provided.

·        The chemotaxonomic analysis of Phyllobacterium onerii new species must be determinate and also they were compared with the Phyllobacterium closed species. Or the author needs to provide an in-silico analysis.

·       The authors needs improve the description paragraphs of Phyllobacterium onerii.

Minor points

·       Line 97, please write the correct formula of the compounds.

Author Response

Dear Reviewer

We would like to thank you for the insightful comments and suggestions. We made all possible changes that were suggested and detailed the changes in the table below. Prior to response  your comment we want to inform  you that all the revisions and improvements are highlighted red in the revised version of our manuscript. We sincerely appreciate your insightful comments on our paper. We would like to thank you again for your valuable time and insight to strengthen our paper.

Yours truly,

Corresponding author on behalf of the authors.

Reviewer 2 Report

Comments and Suggestions for Authors

The article “Genomic insights into the symbiotic and plant growth-promoting traits of Phyllobacterium onerii sp. nov isolated from endemic Astragalus flavescens” authored by EroÄŸlu et al. is devoted to the description of a new species of nodule bacteria isolated from the roots of Astragalus. The authors isolated a pure culture, sequenced the genome and carried out a brief analysis of it, examined some phenotypic properties of the new strain, constructed phylogenetic trees on the basis of 3 different genes and using whole genome-based GBDP distances, and also looked at the ANI and dDDH values of the new strain with its closest phylogenetic relatives. Based on the data obtained, the authors conclude that the isolated strain is classified as a new species. However, there is no information about deposition the strain into international collections of microorganisms, and there are no certificates in the supplements. According to the International Code of Prokaryotic Nomenclature, a new species of bacteria can be described only if it is deposited in at least two international collections located in two different countries. Also, the authors did not conduct chemotaxonomic analysis, which is also mandatory for describing a new species.

There are also a number of minor comments on the text of the article.

1. Everywhere in the text: when the authors refer to a strain, it is necessary to indicate its number in international collections of microorganisms, and not the number in GenBank.

2. L.3. dot should be written after nov

3. L.33-36. It is not necessary to list all 16 species. It is enough just to give a link to LPSN web site

4. L.42. It is better to say that rhizobial phylogenies based on recA correspond with the SSU phylogenies, because special phylogenetic markers are used for every bacterial group

5. L. 110. There is an extra dot here.

6. Section 2.3. What model of microscope did you use?

7. L.131. de novo should be written in italic

8. L.161-165. This information is for methods, not results.

9. Table 1. Were the data for P. trifolii PETP02T and P. bourgognense STM201T also obtained by the authors of the article? If so, why is there are no data for sucrose utilization by P. bourgognense STM201T? If not, you need to cite the work in which these data were obtained.

10. Table 2. Here you can add a comparison with other genomes from this genus.

11. L.188-191. Almost 50 genomes from the genus Phyllobacterium are publicly available. This makes it possible to recalculate the GC composition based on genomic data. This is much more accurate than Tm

12. Table 4. For genes, it is better to indicate not the coordinates of the beginning and end, but the GenBank ID

13. The names of bacterial proteins are written in capital letters and not in italics, and the names of genes are written in small letters and in italics. Check everywhere in the text.

Author Response

Dear Reviewer,

We would like to thank you for the insightful comments and suggestions. We made all possible changes that were suggested and detailed the changes in the table below. Prior to response  your comment we want to inform  you that all the revisions and improvements are highlighted red in the revised version of our manuscript. We sincerely appreciate your insightful comments on our paper. We would like to thank you again for your valuable time and insight to strengthen our paper.

Yours truly,

Corresponding author on behalf of the authors.

Reviewer 3 Report

Comments and Suggestions for Authors

L16: what are ANI and dDDH? The abstract should be understandable to the layman.

L33-36: I think this section would be much more attractive in the form of a table along with the source from which the strain was isolated.

L82: the inclusion of a photograph of Astragalus flavescens would be very valuable. Also nodules.

L97: the subscript is missing at HgCl2

L94-103: is there a methodology, a reference for this procedure?

L123: and were other substrates tested?

L126: please also include here the deposition of the sequence in the base (L296-297),

L150-156: this section is unnecessary; it is part of the methodology.

L173: maybe attach a photo?

L255-263: this is more of a summary, which should be lower in the 'conclusions' section.

Figure 3: the names of the bacteria are not very readable, need to work on the font.

Figure 4: a figure of better quality should be included.

Figure 5: bacterial names are not very legible, need to work on font.

Table 1, 2, 3, 4: please format the table according to the journal guidelines.

Please check that references are formatted according to journal requirements. In addition, I suggest citing more new papers. 21 references are older than 2010, which is 37.5% of all publications. Of these, 7 papers are below 2000. A further 10 references are from 2010-2015 (17.85%). This results in as much as 55.36% of the publications being older than the last 8 years. In my opinion, this is too many. Only 20 papers are from the last 5 years (i.e. from 2018 onwards).

Comments on the Quality of English Language

I have no comments.

Author Response

(The authors gave the same response as above.)

Round 2

Reviewer 1 Report

Comments and Suggestions for Authors

Dear Authors,

The manuscript microorganisms-2816636R-1 entitle “Genomic insights into the symbiotic and plant growth-promoting traits of Phyllobacterium onerii sp. nov isolated from endemic Astragalus flavescens”. The manuscript shows the phenotypic and some genomics characteristics, however, before that was accept it the authors needs considered the following comments.

 The chemotaxonomic analysis of Phyllobacterium onerii new species must be determinate and also they were compared with the Phyllobacterium closed species. Or the author needs to provide an in-silico analysis.

The authors needs improve the description paragraphs of Phyllobacterium onerii, add the chemotaxonomic information.

Author Response

Dear Reviewer

We express our gratitude for your insightful comments and suggestions. We have implemented all feasible changes as per your recommendations and documented these modifications in the table provided below. Your thoughtful feedback on our paper is sincerely valued, and we would like to extend our thanks once more for your precious time and insightful contributions, which have significantly strengthened our work.

Yours truly,

Corresponding author on behalf of the authors.

Reviewer 2 Report

Comments and Suggestions for Authors

Dear authors!

You have corrected the manuscript significantly. I think that the paper can be published in the present form after getting the certificates from international collections. If you will have difficulties with it, you can describe your new species as Candidatus, which in my mind will not reduce the level of your work.

Author Response

(The authors gave the same response as above.)
